# Longitudinal Dyadic Associations between Depressive Symptoms and Life Satisfaction among Chinese Married Couples and the Moderating Effect of Within-Dyad Age Discrepancy

**DOI:** 10.3390/ijerph192013277

**Published:** 2022-10-14

**Authors:** Xiamei Guo

**Affiliations:** School of Public Policy, Xiamen University, Siming District, Xiamen 361005, China; guoxm@xmu.edu.cn

**Keywords:** life satisfaction, depressive symptoms, age discrepancy, longitudinal dyadic influence

## Abstract

Family systems theory defines the family unit as a complex social system in which individual members influence and are influenced by each other. The current study aimed to investigate the longitudinal dyadic associations between life satisfaction and depressive symptoms among a sample of Chinese married couples and the moderating effect of within-dyad age discrepancy. The current sample included 5773 married couples who completed three waves of assessments of the China Family Panel Studies (CFPS) in 2012, 2016, and 2018. The sample was categorized into two groups based on the within-dyad age discrepancy: the younger-wife–older-husband dyads (*n* = 4280, 74.13%) and the older-wife–younger-husband dyads (*n* = 1493, 25.86%). The longitudinal actor–partner interdependence models with multiple-group analysis were used to analyze the data. The results showed that a majority of actor effects were significant across time, and the two groups exhibited the same pattern among the actor effects. The older-wife–younger-husband dyads showed fewer significant partner effects than the younger-wife–older-husband dyads, and most partner effects indicated mutual rather than unidirectional influence. These findings yielded support for the statement of family systems theory that family members interconnect and that the development of one’s well-being needs to be understood in the context of the spouse’s well-being.

## 1. Introduction

A majority of previous studies on the benefits and costs of marriage were carried out in Western countries, with the individual as the unit of analysis [1,2]. When combined with longitudinal designs, research employing the dyad as the unit of analysis could provide valuable insight into how one influences and is influenced by one’s partner. With the goal of expanding the current knowledge about longitudinal dyadic influence among married couples, the current study investigated the actor effects and the partner effects between life satisfaction and depressive symptoms over a 6-year period with a sample of Chinese married couples. Furthermore, the impact of the within-dyad age discrepancy in these dyadic relationships between husbands and wives was also examined in the current study.

### 1.1. Dyadic Influence on Life Satisfaction and Depressive Symptoms among Married Couples

According to family systems theory [3], the family system functions as an interdependent and organized entity. That is, individuals, dyads, or other subsystems within the family system are influencing and being influenced by one another. Changes in one individual within a family are likely to initiate changes in the entire family system, and this might then have an impact on other family members. Therefore, family systems theory focuses primarily on the interaction between family members. Additionally, according to Larson and Almeida, families are crucial settings for daily emotional exchanges [4], and the interpersonal influence among family members occurs through daily interaction. Consequently, it is necessary to expand the scope of inquiry beyond the individual to better understand systemic influence [5,6].

Research has been carried out to examine the interdependence between married couples’ depressive symptoms and life satisfaction, respectively. Several studies have reported unidirectional husband-driven effects, such as the husband’s higher levels of depressive symptoms predicted subsequent increases in depressive symptoms and decreases in life satisfaction in the wife [7,8]. Several studies reported unidirectional wife-driven effects, as well. For example, King et al. found a relationship between the wife’s life satisfaction at a previous time and the husband’s life satisfaction at a later time [9]. Additionally, depressive symptoms in the wife affected the husband’s depressive symptoms, but not the other way around [10]. A pattern of mutual relationships has been noted in some research in addition to these unidirectional husband-driven or wife-driven tendencies [11,12]. Significant actor and partner effects for relationship satisfaction on depressive symptoms have also been reported among same-sex couples [13].

Researchers have offered various explanations for these contradictory results. On the one hand, women may be more relationally dependent than males, according to Cross and Madson [14]. Larson and Almeida also suggested that the husband is more likely to be the one who directs the family’s emotional flow than the woman [4]. As a result, husbands’ depressive symptoms may affect wives’ mental health and level of happiness more so than the other way around, which may result in the unidirectional husband-driven effect. On the other hand, the wife-driven impact may be related to the fact that depressive symptoms are more common in women than men in the general population, as suggested by Kessler and colleagues [15]. Thomeer, Umberson, and Purdrovska found that the spouse of depressive women tended to exhibit signs of incapability of dealing with emotions [16]. This might result in the familial dynamics becoming more hostile and isolated, which might eventually lead to the wife-driven transmission of depressive symptoms. When considered collectively, the contradictory results in the literature highlight the need for more research on the long-term links between depressive symptoms and life satisfaction in married couples.

Besides the dyadic associations, the interrelationships between depressive symptoms and life satisfaction at the individual level should also be taken into account. Depressive symptoms were conceptualized as a cross-sectional or longitudinal indicator of life satisfaction in some research [17], while other studies found that life satisfaction was associated with depressive symptoms at a later time [18,19]. While analyzing the between-individual influence among married couples, it would be desirable to account for longitudinal within-individual sequences with both variables. For instance, Downward et al. reported that there might be a potential vicious circle between reduced mental health and lower relationship satisfaction among males, whereas for females, it was more of a unidirectional influence from reduced relationship satisfaction to poor mental health [20]. In addition, several factors have been linked to both depressive symptoms and life satisfaction among married couples, such as levels of education, physical health, financial stress, history of childhood abuse, substance use and abuse, social support, infertility, etc. [19,21,22,23].

### 1.2. The Effect of Within-Dyad Age Discrepancy on Life Satisfaction and Depressive Symptoms

The within-dyad age discrepancy might be a straightforward yet essential factor in figuring out the dyadic influence between spouses. From the evolutionary psychology point of view, males often prefer younger mates, whereas females generally prefer older partners. This tendency is related to the traditional gender roles in society. That is, women carry the burden of bearing and raising children, while males are responsible for working and supporting the family. Based on the evolutionary perspective, marriages with a younger wife and an older husband might be more fitted to the social norms and expectations, leading to higher levels of life satisfaction and lower depressive symptoms than couples with an older wife and a younger husband.

This theoretical viewpoint is supported by several empirical research. According to Groot and Van Den Brink’s findings, both spouses were happier when the husband was older than his wife than their counterparts among which the husband was younger than his wife [24]. Drefahl reported that women who marry younger men may pass away sooner than those who marry older men, whereas men who marry younger women live longer [25]. In addition, Kim, Park, and Lee found that the older-wife–younger-husband dyads exhibited increased severity in depressive symptoms among both husbands and wives over time than the younger-wife–older-husband dyads [26]. These findings shed some light on the potential consequences of within-dyad age discrepancy, but they did not use the dyadic analysis approach to evaluate actor and partner effects independently. In addition, it is still unclear whether previous findings could be generalized to Chinese spouses, given that a majority of earlier studies were conducted in Western nations. Eastern and Western cultural contexts may result in differential couple dynamics, because the social norms and expectations of husbands and wives differed to some degree. For example, eastern culture rooted in collectivism praises scarification of personal needs for the sake of relational stability, whereas Western culture rooted in individualism does not. Therefore, it is of importance to examine couple dynamics with samples from non-Western cultural context in order to expand the current knowledge. It should be mentioned that, in traditional patriarchal Chinese culture, the younger-wife–older-husband combo is preferable, probably to a greater degree than it is in other countries in the world. It is even subtly reflected in the fact that the legal marriage age in China is 20 for women, while it is 22 for men, while in most countries the legal marriage age limit was the same regardless of gender. Therefore, one may anticipate that Chinese couples with older husbands and younger wives would fare better in terms of their relationships and personal lives than couples with younger husbands and older wives.

### 1.3. The Current Study

Based on the previous findings in the literature, one can conclude that the longitudinal interdependence and dyadic influence between husbands and wives is worth further investigation. It would be of interest to probe whether within-dyad age discrepancy would affect the within-person and between-person associations of depressive symptoms and life satisfaction (as shown in Figure 1). However, the majority of earlier research used samples from Western culture [7,8,9,16,18], whereas just a few were conducted with non-Western samples [19,27]. Therefore, the current study tested the following hypotheses by using married couples’ data from three waves of a national representative dataset with Chinese families: (1) the husbands’ and the wives’ life satisfaction and depressive symptoms at an earlier assessment would be significantly associated with their own life satisfaction and depressive symptoms at the later assessment (the actor effects); (2) husbands’ and wives’ life satisfaction and depressive symptoms at an earlier assessment would be significantly associated with their spouse’s life satisfaction and depressive symptoms at the later assessment (the partner effects); (3) dyads with a younger wife and an older husband would show higher levels of life satisfaction and lower levels of depressive symptoms than dyads with an older wife and a younger husband across time; and (4) the two groups based on within-dyad age discrepancy would exhibit different dyadic interdependence patterns in depressive symptoms and life satisfaction across time. Age, physical health, income, and marital satisfaction were used as control variables in this study given the evidence that these variables might be correlated with depressive symptoms and life satisfaction among couples [28,29,30].

## 2. Method

### 2.1. Participants

The current study used secondary data from China Family Panel Studies (CFPS). CFPS is an ongoing biennial longitudinal social survey project of Chinese communities, families, and individuals that was launched in 2010 [31]. To ensure national representability, the CFPS baseline sample was drawn by using a multi-stage (county, village, and household) probability stratification. CFPS includes individual and family measurements, which encompass areas such as the individual’s physical and mental health, economic activities, educational outcomes, and migrations, as well as family relationships. The follow-up assessments have been undertaken every other year. The latest wave of data released was the 2018 follow-up assessment. CFPS used two different measurements for depressive symptoms across follow-ups. One measurement was administered at baseline and during the 2014 follow-up, and the other one was administered during the 2012, 2016, and 2018 follow-ups. To incorporate the most waves and keep the measurement equivalence across assessments, the current study used the data from 2012, 2016, and 2018 follow-ups. The sample selection process was presented a flowchart (Figure 2). Based on the birth year and birth month information, a majority of the sample (*n* = 4280, 72.8%) were dyads in which the wife was the younger one. About one-quarter of the sample (*n* = 1493, 25.4%) consisted of dyads in which the wife was the older one. The demographic characteristics of the final sample are presented in Table 1.

### 2.2. Measurement

Depressive symptoms were assessed by a 20-item version of the Center for Epidemiological Studies Depression Scale (CES-D) in the 2012 and 2016 assessments [32]. An 8-item version of the CES-D was used in the 2018 assessment. To make sure the measurement was comparable across waves, this study utilized the same 8 items of the CES-D in the 2012, 2016, and 2018 assessments to calculate the levels of depressive symptoms. Participants answered the frequency of feeling depressed, happy (reversed item), lonely, sad, having trouble doing anything, having sleeping problems, living lively (reversed item), and feeling that it would be difficult to keep on living during the previous week. Each item was rated on a 4-point Likert scale, with higher scores indicating higher frequencies. The sum of these eight items was calculated to represent the levels of depressive symptoms, and the Cronbach’s alpha for CES-D ranged from 0.76 to 0.78 across three waves.

One broad question was used to assess the level of life satisfaction: “How satisfied were you with your life?” Participants rated their satisfaction on a 5-point Likert scale, with higher scores indicating higher life satisfaction.

Based on the information of birth year and month in the demographic section, dyads were categorized into two groups (0 = the wife was younger than her husband; 1 = the wife was older than her husband). Covariates included age, self-reported physical health, and annual income in the 2012 assessment, and marital satisfaction measured in the 2014 assessment. Physical health and marital satisfaction were rated on a 5-point Likert scale, with higher scores indicating more physical health problems or higher levels of satisfaction.

### 2.3. Analytic Procedures

Both a missing-data analysis and descriptive analysis were performed with the SPSS version 22. The reciprocal effects between husbands and wives across time were analyzed by the Actor–Partner Interdependence Model (APIM) [33] with the AMOS 24.0 (see Figure 3 for the full model). The APIM is a path analysis that could examine the actor effects and the partner effects simultaneously to account for the statistical interdependence at the dyadic level. The actor effect describes how a person’s score on a predictor variable relates to that same person’s score on an outcome variable, whereas the partner effect shows how a person’s score on a predictor variable influences his or her spouse’s score on an outcome variable [33]. In the current analysis, an APIM with the actor and partner effects of depressive symptoms and life satisfaction across three waves was analyzed. The covariate variables included annual income, age, and marital satisfaction and were correlated to the dependent variables in 2018. The goodness-of-fit of the model was evaluated by the Root Mean Square Error of Approximation (RMSEA), the Tucker–Lewis Index (TLI), and the Comparative Fit Index (CFI). RMSEA values of less than 0.06 and CFI and TLI values of greater than 0.95 indicate a good model fit [34]. The moderating effect of the within-dyad age discrepancy (0 = dyads in which the wife was younger than the husband; 1 = dyads in which the wife was older than the husband) was tested with the multiple-group analysis. An unconstrained model would be compared to models with equal structural weights, intercepts, means, covariance, etc. The fit of the nested models was compared with chi-square tests. If the unconstrained model fit the data better than models with equal constraints, then the two groups were considered to be different in terms of the relationships among the variables.

## 3. Results

Independent-sample *t*-tests were used to compare the differences in life satisfaction, depressive symptoms, and age in 2012 between the 5773 dyads used in the current study and the remaining dyads included in the CFPS but not included in the current study. The results showed that dyads in the current study were younger [*t*(9780.24) = 11.34, *p* < 0.01; *t*(10549.13) = 11.90, *p* < 0.01] than the dyads which were not included in the current study. They did not differ on the levels of life satisfaction in 2012, but those included in the current study also exhibited significantly lower levels of depressive symptoms in 2012 [*t*(7875.20) = 5.19, *p* < 0.01; *t*(7300.65) = 6.15, *p* < 0.01]. Little’s MCAR test was conducted to examine the mechanism of missing data in the current sample. The results showed that the current data were not missing completely at random [*χ*^2^(2593) =3381.31, *p* < 0.05].

The means and standard deviations of all the variables based on the within-couple age-discrepancy grouping information were presented in Table 1. Between-group differences in their depressive symptoms and life satisfaction in each assessment were analyzed by independent-sample *t*-tests. The only significant difference between the two groups was that husbands who were older than their wives exhibited higher levels of depressive symptoms in 2012 [*t*(5251) = 3.08, *p* < 0.01] than their counterparts who were younger than their wives. Therefore, Hypothesis 3 was not supported. Within-dyad differences were examined by paired-sample *t*-tests for each group separately. The results showed that the wives in both groups reported higher levels of depressive symptoms than their husbands in all three assessments (all *p*-values < 0.001). Life satisfaction among the wives in the younger-wife–older-husband group was higher than that of their husbands in 2012 and 2016 [*t*(3749) = 2.01, *p* < 0.05; *t*(2819) = 4.35, *p* < 0.001], whereas wives and husbands did not differ in their life satisfaction in the older-wife–younger-husband group across time (all *p*-values > 0.05). The two groups also showed significant differences in some covariates. The husbands in the younger-wife–older-husband group were older than their counterparts in the older-wife–younger-husband group [*t*(2843.63) = 8.81, *p* < 0.001], whereas their wives were younger than the wives in the other group [*t*(2784.29) = −4.54, *p* < 0.001]. The older-wife–younger-husband group displayed higher levels of marital satisfaction for both husbands [*t*(2610.99) = −2.86, *p* < 0.001] and wives [*t*(2596.15) = −3.40, *p* < 0.01] than the younger-wife–older-husband group.

Correlation analysis showed that one’s depressive symptoms and life satisfaction were significantly and negatively correlated (all *p*-values < 0.01) across three assessments (Table 2). Husbands’ and wives’ ratings of depressive symptoms and life satisfaction were significantly correlated both cross-sectionally and longitudinally (all *p*-values < 0.05). In addition, higher income and higher marital satisfaction were significantly correlated with lower levels of depressive symptoms and higher levels of life satisfaction across time. Age was positively related to both life satisfaction across three waves and depressive symptoms in 2012 and 2016 (all *p*-values < 0.05). Significant negative correlations were found between age and depressive symptoms in 2018 (all *p*-values < 0.05).

### Actor–Partner Interdependence Model Analysis

The full unconstrained model exhibited a good fit to the data (CFI = 0.98, TLI = 0.94, RMSEA = 0.031, 90% CI: [0.029, 0.034]). The multiple-group analysis showed that the model with equal structural weights fit the data significantly worse than an unconstrained model [*χ*^2^(44) =61.04, *p* < 0.05]. Therefore, the unconstrained model was retained as the final model, which indicated that the parameter estimates between the two groups were not equal. Hypothesis 4, regarding the moderating effect of the within-dyad age difference, was supported. The results of the parameter estimates are presented in Table 3.

Significant and positive actor effects were found between one’s depressive symptoms in 2012 and 2016 and between those assessed in 2016 and 2018 (all *p*-values < 0.05). One’s life satisfaction at an earlier assessment was also positively correlated with his or her life satisfaction at the later assessment (all *p*-values < 0.05). The cross-lagged associations between one’s life satisfaction and depressive symptoms were all significant and negative from 2012 to 2016 (all *p*-values < 0.05). The paths from one’s depressive symptoms in 2016 to the same person’s life satisfaction in 2018 were not significant (both *p*-values > 0.05), whereas positive associations were found between one’s life satisfaction in 2016 and depressive symptoms in 2018 (both *p*-values < 0.05). Taking these results together, the first hypothesis of the current study regarding the significant actor effect was mostly supported. In addition, the older-husband–younger-wife dyads and the younger-husband–older-wife dyads exhibited the same significant paths of actor effects.

Among the paths representing partner effects, the two groups showed several differences. Some partner effects were significant among both groups, whereas some other partner effects were significant only among the younger-wife–older-husband group. From 2012 to 2016, the cross-lagged associations between wives’ depressive symptoms and husbands’ depressive symptoms and the path from husbands’ life satisfaction to wives’ life satisfaction were significant and positive for both groups (all *p*-values < 0.05). From 2016 to 2018, the paths from husbands’ depressive symptoms to wives’ depressive symptoms and life satisfaction were both negative across the two groups (all *p*-values < 0.05). In addition, the path from wives’ life satisfaction in 2016 to husbands’ depressive symptoms in 2018 was significantly negative (*p*-values < 0.05).

For those younger-wife–older-husband dyads, a few more significant partner effects were found. Only four out of the sixteen partner effects were not statistically significant among this group. The non-significant paths included the two paths from one’s life satisfaction in 2012 to the spouse’s depressive symptoms in 2016, the path from wives’ depressive symptoms in 2012 to husbands’ life satisfaction in 2016, and the path from husbands’ life satisfaction in 2016 to wives’ life satisfaction in 2018. Given the findings of the significant partner effects in both groups, Hypothesis 2 was partly supported.

Several significant associations between the covariates and dependent variables in 2018 were found. Wives’ age was positively related to their life satisfaction (*p*-values < 0.05) in both groups but not depressive symptoms in either group (*p*-values > 0.05). Husbands’ age was also positively related to their life satisfaction in 2018 in both groups (*p*-values < 0.05). Wives’ marital satisfaction in 2014 was positively related to their life satisfaction and negatively related to their depressive symptoms in 2018 across both groups (*p*-values < 0.05). Husbands’ marital satisfaction in 2014 was negatively related to their depressive symptoms and positively related to their life satisfaction in 2018 among those younger-wife–older-husband dyads (*p*-values < 0.05). A negative association was found between husbands’ age and depressive symptoms in 2018 among the older-wife–younger-husband dyads (*p* < 0.05). Wives’ income in 2012 was negatively related to their depressive symptoms in 2018 in both groups and was negatively related to their life satisfaction in 2018 among the younger-wife–older-husband dyads (*p*-values < 0.05). Husbands’ income in 2012 was only negatively related to depressive symptoms among the younger-wife–older-husband group (*p* > 0.05).

## 4. Discussion

The primary goal of this study was to investigate the long-term dyadic influence between husbands and wives on their depressive symptoms and life satisfaction, as well as the moderating effect of the within-dyad age difference, using data from a sample of Chinese married couples across 6 years. The overall findings on the actor and partner effects supported the first two hypotheses. In most scenarios, the younger-wife–older-husband dyads and the older-wife–younger-husband dyads did not differ in terms of their depressive symptoms and life satisfaction, which did not support Hypothesis 3. The multiple-group comparison analysis revealed that the dyadic associations between life satisfaction and depressive symptoms differed between the younger-wife–older-husband dyads and the older-wife–younger-husband dyads across three assessments. Specifically, the younger-wife–older-husband dyads exhibited fewer significant partner effects than the older-wife–younger-husband dyads. Therefore, Hypothesis 4 was partially supported.

The current study found a relationship between a person’s depressive symptoms and life satisfaction throughout the course of six years. These actor effects indicated that these two constructs were relatively stable among Chinese people. In addition, there was a negative long-term association between one’s depressive symptoms and life satisfaction. Prior research suggested that individuals with greater depressive mood responses to daily marital discord exhibited greater long-term increases in depressed mood and marital risk [35]. The current findings regarding the cross-lagged associations between depressive symptoms and life satisfaction added to the body of literature regarding the interrelations between these two constructs [17,18]. The current findings showed that one’s depressive symptoms and life satisfaction exhibited bidirectional associations between 2012 and 2016. From 2016 to 2018, one’s depressive symptoms were no longer significantly related to his or her life satisfaction prospectively. The improvement in household living standards and the social environment may have made up for the detrimental impact of depressive symptoms on life satisfaction, as the Chinese economy grew consistently from 2016 to 2018. To acquire more evidence about the sequential influence, the future study may examine the correlations between depressive symptoms and life satisfaction by using more data points [9].

The actor effects of the two groups of couples were similar, but their partner effects varied to some extent, as was predicted. Nearly all cross-lagged partner effects among the younger-wife–older-husband dyads were statistically significant. Only two routes were insignificant: the one between husbands’ life satisfaction in 2016 and wives’ life satisfaction in 2018, and the one between wives’ depressive symptoms in 2012 and husbands’ life satisfaction in 2016. In the younger-wife–older-husband dyads, the husband and wife were likely to contribute equally to the partner effects. These results were consistent with previous research which found a reciprocal rather than a unilateral impact between both partners, and that the magnitude of effects was equivalent [12,36,37]. They proposed that the spousal similarity in coping might be related to the mutual associations of depressive symptoms. The current findings regarding the younger-wife–older-husband dyads, which represent the majority of Chinese married couples, supported the notion of interdependence between family members from the family systems theory [3]. Therefore, individual behavioral and psychological health could be better understood in the context of the family as a whole.

Among the older-wife–younger-husband dyads, fewer significant partner effects were found. The positive relationships between one’s depressive symptoms in 2012 and the spouse’s depressive symptoms in 2016, as well as the positive relationships between one’s life satisfaction in 2012 and the spouse’s life satisfaction in 2016, were significant among these couples. In addition, wives’ depressive symptoms and life satisfaction were both negatively associated with husbands’ depressive symptoms in 2018, whereas husbands’ depressive symptoms in 2016 were negatively related to wives’ life satisfaction in 2018. These findings imply that the depressive symptoms of the husbands in these older-wife–younger-husband dyads might depend more on their wives over time than the other way around. These findings are in line with previous research regarding the wife-driven influence on husbands’ depressive symptoms [9,10].

The current study discovered that the two groups did not differ on most measurements of life satisfaction and depressive symptoms and that the older-wife–younger-husband group reported better levels of marriage satisfaction than their counterparts. These results challenged earlier research that claimed that having an older wife would be harmful to both partners’ well-being [25,26]. In contrast to the younger-wife–older-husband couples, the older-wife–younger-husband couples exhibited fewer partner effects. When combined, these results may show a pattern resembling that obtained by Kouros and Cummings [8], who hypothesized that depressive symptoms would be more strongly influenced by the partner in marriages with lower marital satisfaction. Nevertheless, the current data suggest that the within-dyad age discrepancy may be related to couple dynamics and that the non-normative combination of an older wife and younger husband may even be somewhat associated with better outcomes. A recent meta-analysis on marital satisfaction of Chinese couples revealed that husbands’ marital satisfaction exhibited almost no change in the past few decades, but wives’ marital satisfaction showed an obvious trend of decreasing [38]. The current findings might be useful in dispelling some social stigmas associated with the older-wife–younger-husband coupling as China’s conventional patriarchal family relations experience a fast transformation.

It is important to be aware of the current study’s limitations. First, the current sample was not missing completely at random, and the generalizability of the current findings was restricted. Longitudinal methods that can handle missing data, such as dyadic latent difference models, could be used in future research. Second, because the current study used only three assessment points, the possibility of testing the nonlinear effect among variables of interest was ruled out. Third, the average level of depressive symptoms of the current sample was relatively low, in that the data were from a nationally representative dataset. Couples with a depressed partner might exhibit different dyadic influence patterns on depressive symptoms and life satisfaction, and the current findings might not be generalizable to them. However, the CFPS dataset did not include diagnostic information regarding depressive disorder, thus precluding the possibility of probing the differences between a clinical subsample from a general sample of Chinese married couples. Fourth, several factors associated with depressive symptoms and life satisfaction were not accounted for in the current study, such as physical health, spousal infertility, house stability, chronic stress, sexual satisfaction, etc. Last but not least, while the current study investigated interpersonal influence by using the marital dyad as the unit of analysis, children are also important influential members of the family system. Given the evidence linking marital satisfaction with the parent–child relationship [39], future research using a triadic unit of analysis may provide more insight into the broader picture of family dynamics.

## 5. Conclusions

Depressive symptoms and life satisfaction among Chinese married couples are stable and longitudinally interrelated at the individual level. The findings regarding the partner effects are in line with the family systems theory, which holds that individuals are influencing and influenced by their family members, hereby their spouse. The current research suggests that Chinese couples with an older wife and a younger husband might also gain in psychological well-being, even though dyads with a young wife and an older husband may have higher reproduction values from an evolutionary psychological standpoint and are a better fit for the social norm. These discoveries contributed to the married-couple dyadic studies, the majority of which has been carried out in the West. Future research investigating the underlying mechanisms of interpersonal influence within the context of family is still needed.

## Figures and Tables

**Figure 1 ijerph-19-13277-f001:**
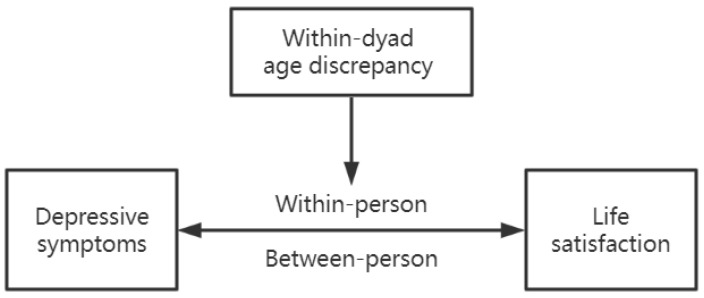
The conceptual model of associations between variables.

**Figure 2 ijerph-19-13277-f002:**
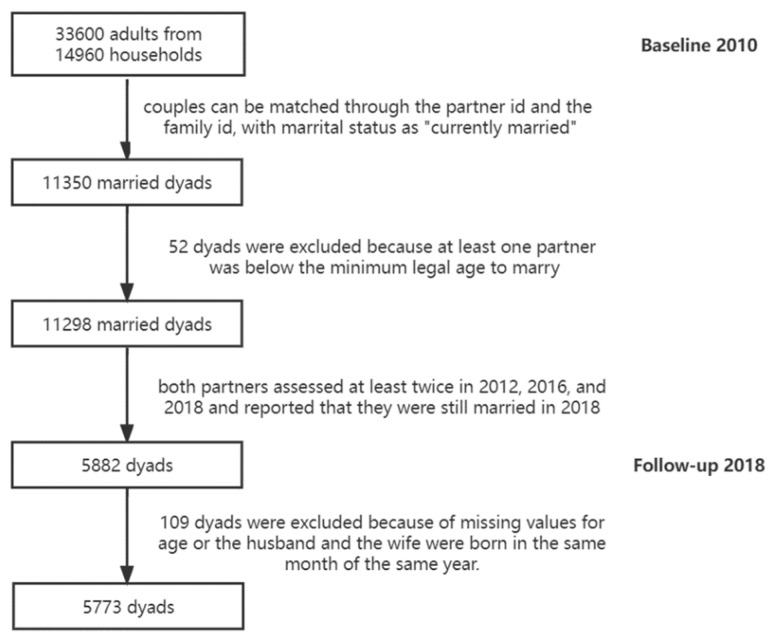
The flowchart of sample selection process.

**Figure 3 ijerph-19-13277-f003:**
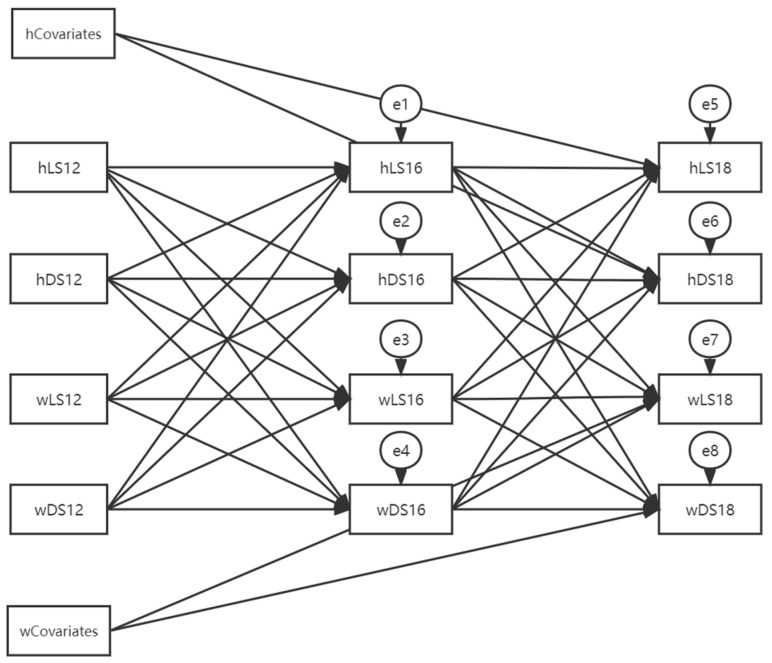
The full APIM model. Note: w, wives’; h, husbands’; MS, marital satisfaction; LS, life satisfaction; DS, depressive symptoms. Covariates include age and income (log-transformed) in 2012 and marital satisfaction in 2014. DS and LS measured in 2012 and the covariates were all set to be covaried. Several pairs of residual errors were set to be intercorrelated: e1 and e3, e2 and e4, e5 and e7, e6 and e8, e1 and e5, e2 and e6, e3 and e7, and e4 and e8. These covariance paths were omitted from the path diagram for the ease of reading.

**Table 1 ijerph-19-13277-t001:** Demographic characteristics and descriptive analysis.

Variable	Younger-Wife–Older-Husband (*n* = 4280)			Older-Wife–Younger-Husband (*n* = 1493)		
	Range	Mean	S.D.	Range	Mean	S.D.
*Covariates*						
wage12	22–78	46.94	11.71	24–78	48.45	10.86
hage12	24–87	50.00	11.91	24–78	47.06	10.77
wincome12	1–5.57	4.00	0.58	0–0.57	3.93	0.63
hincome12	0.30–5.57	4.21	0.47	0–5.95	4.24	0.42
wMS14	1–5	4.57	0.80	1–5	4.64	0.72
hMS14	1–5	4.38	0.95	1–5	4.48	0.86
wLS12	1–5	3.37	1.05	1–5	3.36	1.03
wLS16	1–5	3.73	1.08	1–5	3.71	1.11
wLS18	1–5	4.12	0.95	1–5	4.11	0.96
hLS12	1–5	3.33	1.03	1–5	3.36	1.05
hLS16	1–5	3.65	1.06	1–5	3.65	1.04
hLS18	1–5	4.11	0.93	1–5	4.09	0.95
wDS12	0–24	5.55	3.90	0–24	5.37	3.86
wDS16	0–24	5.49	4.13	0–24	5.34	4.04
wDS18	0–24	5.97	4.11	0–24	5.75	3.98
hDS12	0–24	4.51	3.50	0–24	4.17	3.47
hDS16	0–24	4.39	3.71	0–24	4.22	3.65
hDS18	0–24	4.85	3.83	0–24	4.67	3.72

Note: w, wives’; h, husbands’; LS, life satisfaction; DS, depressive symptoms; MS, marital satisfaction.

**Table 2 ijerph-19-13277-t002:** Correlation coefficients between variables.

Variable		(1)	(2)	(3)	(4)	(5)	(6)	(7)	(8)	(9)	(10)	(11)	(12)	(13)	(14)	(15)	(16)	(17)
(1)	wLS12	1																
(2)	wLS16	0.28 **	1															
(3)	wLS18	0.21 **	0.30 **	1														
(4)	hLS12	0.29 **	0.18 **	0.13 **	1													
(5)	hLS16	0.14 **	0.28 **	0.16 **	0.29 **	1												
(6)	hLS18	0.10 **	0.14 **	0.20 **	0.22 **	0.33 **	1											
(7)	wDS12	−0.28 **	−0.16 **	−0.14 **	−0.15 **	−0.08 **	−0.07 **	1										
(8)	wDS16	−0.18 **	−0.27 **	−0.17 **	−0.11 **	−0.16 **	−0.09 **	0.40 **	1									
(9)	wDS18	−0.20 **	−0.19 **	−0.24 **	−0.1 **	−0.12 **	−0.12 **	0.41 **	0.50 **	1								
(10)	hDS12	−0.13 **	−0.11 **	−0.08 **	−0.30 **	−0.19 **	−0.16 **	0.36 **	0.20 **	0.20 **	1							
(11)	hDS16	−0.10 **	−0.17 **	−0.11 **	−0.21 **	−0.33 **	−0.21 **	0.21 **	0.34 **	0.23 **	0.41 **	1						
(12)	hDS18	−0.11 **	−0.14 **	−0.13 **	−0.20 **	−0.22 **	−0.27 **	0.19 **	0.22 **	0.31 **	0.41 **	0.48 **	1					
(13)	wage	0.04 **	0.12 **	0.11 **	0.11 **	0.17 **	0.11 **	0.08 **	0.07 **	0.05 **	0.005	−0.020	−0.046 **	1				
(14)	hage	0.030 *	0.113 **	0.105 **	0.103 **	0.165 **	0.108 **	0.094 **	0.082 **	0.063 **	0.027 *	−0.01	−0.03 *	0.97 **	1			
(15)	wMS	0.17 **	0.17 **	0.20 **	0.09 **	0.09 **	0.09 **	−0.15 **	−0.15 **	−0.14 **	−0.09 **	−0.09 **	−0.10 **	0.01	−0.003	1		
(16)	hMS	0.08 **	0.08 **	0.08 **	0.09 **	0.14 **	0.15 **	−0.08 **	−0.08 **	−0.08 **	−0.14 **	−0.13 **	−0.12 **	0.01	−0.01	0.25 **	1	
(17)	wincome	0.11 **	0.05 *	−0.01	0.06 *	0.05 *	0.03	−0.22 **	−0.17 **	−0.19 **	−0.15 **	−0.13 **	−0.13 **	−0.01	−0.004	0.04	0.03	1
(18)	hincome	0.05 **	0.02	−0.02	0.08 **	0.04 *	0.02	−0.15 **	−0.11 **	−0.13 **	−0.18 **	−0.12 **	−0.13 **	−0.08 **	−0.06 **	0.03	0.07 **	0.49 **

Note: w, wives’; h, husbands’; LS, life satisfaction; DS, depressive symptoms; MS, marital satisfaction; ** *p* < 0.01; * *p* < 0.05.

**Table 3 ijerph-19-13277-t003:** APIM analysis results.

			Younger-Wife–Older-Husband (*n* = 4280)				Older-Wife–Younger-Husband (*n* = 1493)			
			Estimate	SE	CR	*p*	Estimate	SE	CR	*p*
*Actor effects*										
wLS12	→	wLS16	0.25	0.02	14.83	***	0.21	0.03	7.55	***
hLS12	→	hLS16	0.26	0.02	14.98	***	0.24	0.03	8.89	***
wLS16	→	wLS18	0.47	0.06	7.93	***	0.70	0.11	6.20	***
hLS16	→	hLS18	0.56	0.06	8.69	***	0.66	0.12	5.39	***
wDS12	→	wDS16	0.39	0.02	23.50	***	0.34	0.03	11.74	***
hDS12	→	hDS16	0.38	0.02	22.41	***	0.37	0.03	13.56	***
wDS16	→	wDS18	1.00	0.05	19.93	***	0.94	0.10	9.76	***
hDS16	→	hDS18	1.10	0.06	18.81	***	0.97	0.08	11.73	***
wDS12	→	wLS16	−0.02	0.00	−4.59	***	−0.03	0.01	−4.43	***
wLS12	→	wDS16	−0.31	0.05	−5.88	***	−0.27	0.09	−2.89	0.004
wDS16	→	wLS18	−0.01	0.01	−1.08	0.28	0.02	0.01	1.91	0.06
wLS16	→	wDS18	0.22	0.08	2.94	0.003	0.32	0.13	2.53	0.01
hDS12	→	hLS16	−0.04	0.01	−7.24	***	−0.04	0.01	−4.74	***
hLS12	→	hDS16	−0.29	0.05	−5.87	***	−0.53	0.08	−6.63	***
hDS16	→	hLS18	0.00	0.01	0.01	0.99	0.02	0.01	1.31	0.19
hLS16	→	hDS18	0.46	0.09	5.37	***	0.36	0.13	2.84	0.004
*Partner effects*										
wDS12	→	hDS16	0.07	0.01	5.27	***	0.07	0.02	3.23	0.001
hDS12	→	wDS16	0.08	0.02	4.80	***	0.11	0.03	3.58	***
wDS16	→	hDS18	−0.14	0.02	−6.28	***	−0.06	0.03	−1.73	0.08
hDS16	→	wDS18	−0.14	0.03	−5.21	***	−0.12	0.04	−2.86	0.004
wLS12	→	hLS16	0.08	0.02	5.11	***	0.04	0.03	1.40	0.16
hLS12	→	wLS16	0.10	0.02	5.75	***	0.12	0.03	4.59	***
wLS16	→	hLS18	−0.05	0.02	−2.29	0.02	−0.04	0.03	−1.25	0.21
hLS16	→	wLS18	−0.03	0.02	−1.21	0.23	−0.04	0.04	−1.07	0.28
wDS12	→	hLS16	0.00	0.00	0.19	0.85	0.01	0.01	0.93	0.35
hDS12	→	wLS16	−0.01	0.01	−2.08	0.04	0.00	0.01	−0.03	0.98
wDS16	→	hLS18	−0.01	0.00	−2.70	0.01	−0.01	0.01	−1.17	0.24
hDS16	→	wLS18	−0.01	0.01	−2.35	0.02	−0.02	0.01	−3.01	0.003
wLS12	→	hDS16	−0.09	0.05	−1.95	0.05	0.13	0.08	1.67	0.10
hLS12	→	wDS16	−0.08	0.06	−1.38	0.17	−0.13	0.09	−1.38	0.17
wLS16	→	hDS18	−0.33	0.06	−5.89	***	−0.20	0.09	−2.31	0.02
hLS16	→	wDS18	−0.32	0.06	−5.09	***	−0.14	0.10	−1.47	0.14
*Covariates*										
wincome12	→	wLS18	−0.09	0.04	−2.38	0.02	−0.09	0.06	−1.53	0.13
wincome12	→	wDS18	−0.56	0.14	−3.91	***	−0.70	0.22	−3.22	0.001
hincome12	→	hLS18	−0.04	0.04	−1.07	0.29	0.04	0.07	0.53	0.60
hincome12	→	hDS18	−0.32	0.14	−2.22	0.03	−0.49	0.26	−1.89	0.06
hMS14	→	hDS18	−0.16	0.07	−2.42	0.02	0.01	0.12	0.11	0.911
hMS14	→	hLS18	0.13	0.02	7.40	***	0.06	0.03	1.81	0.07
wMS14	→	wDS18	−0.12	0.06	−2.12	0.03	−0.35	0.11	−3.18	0.001
wMS14	→	wLS18	0.15	0.02	9.70	***	0.12	0.03	4.12	***
wage	→	wLS18	0.01	0.00	4.80	***	0.01	0.00	2.91	0.004
wage	→	wDS18	0.01	0.01	1.58	0.12	0.00	0.01	0.25	0.80
hage	→	hDS18	−0.01	0.00	−1.60	0.11	−0.02	0.01	−2.47	0.01
hage	→	hLS18	0.00	0.00	2.73	0.01	0.01	0.00	3.67	***

Note: w, wives’; h, husbands’; MS, marital satisfaction; LS, life satisfaction; DS, depressive symptoms; *** *p* < 0.001.

## Data Availability

CFPS data are publicly available (www.isss.pku.edu.cn/cfps/en/index.htm accessed on 12 October 2022).

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
