# Peer review of "Longitudinal Dyadic Associations between Depressive Symptoms and Life Satisfaction among Chinese Married Couples and the Moderating Effect of Within-Dyad Age Discrepancy"

_ijerph, 2022, doi:10.3390/ijerph192013277_

Round 1

Reviewer 1 Report

This is a well-performed study that provides a relevant contribution to the field. The manuscript shows that dyadic influences between married couple is very important on life satisfaction and depression. 

There are a few concerns to be addressed:

Depressive symptoms of participants is very low. This is limitations of this study. Is this study about dyadic effects of couples on not depressive symptoms?

Author Response

Comment:

Depressive symptoms of participants is very low. This is limitations of this study. Is this study about dyadic effects of couples on not depressive symptoms?

Response:

Reviewer 3 also raised a question related to this comment. The current sample was not from a clinical subpopulation, but from a nationally representative dataset. So, the average level of depressive symptoms was low, as Reviewer 1 pointed out.  I have added this into the limitation section, that the results from the current study might not be generated to couples with a depressed partner.

Reviewer 2 Report

Thank you for inviting me to review this interesting study. This study leveraged a rich dataset to delineate the longitudinal dyadic relations between depressive symptoms and life satisfaction. Likewise, within-dyad age discrepancy was investigated. In general, this manuscript is very well-written and easy to comprehend. Hence, this study has significant conceptual and practical implications and will gather a broad range of readership. I have just listed a few minor suggestions for the author's consideration.

1. In several lines of the manuscript, the author argued that one of the significant gaps in the extant research is the samples on studied associations derived from the west. I would encourage the author to elaborate more specifically on how a study based on the Eastern cultural context, such as in China, would be very important for the under-investigated associations.

2. Under the current study section, I feel that a conceptual model illustrating the research aims at that stage would be helpful for better readability and clarity.

3. A flowchart may better articulate inclusion and exclusion criteria. The current form, although clear and concise, brings potential burdens to read.

4. The terms, in Table 1, may be presented consistently: 2012 or 12.

5. It is great to use different paragraphs for different measures for better readability in the measurement section.

6. The presentation of Figure 1 can be improved, as several lines connected in the figure are hard to read. If residual errors are inter-correlated, could you please add a footnote under the figure for simplicity instead of mapping all in the figure?

Author Response

1. In several lines of the manuscript, the author argued that one of the significant gaps in the extant research is the samples on studied associations derived from the west. I would encourage the author to elaborate more specifically on how a study based on the Eastern cultural context, such as in China, would be very important for the under-investigated associations.

Response: Thanks for the suggestion. We have added several lines regarding the significance of considering cultural context while examining couple dynamics on page 3.

2. Under the current study section, I feel that a conceptual model illustrating the research aims at that stage would be helpful for better readability and clarity.

Response: We added Figure 1 to illustrate the conceptual associations between study variables. Hopefully it could address this comment.

3. A flowchart may better articulate inclusion and exclusion criteria. The current form, although clear and concise, brings potential burdens to read.

Response: Thank you for this suggestion. We have replaced the description by a flowchart.

4. The terms, in Table 1, may be presented consistently: 2012 or 12.

Response: This has been corrected in the revised manuscript. Thanks for the notice.

5. It is great to use different paragraphs for different measures for better readability in the measurement section.

Response: We have changed the layout of measurement section according to this comment.

6. The presentation of Figure 1 can be improved, as several lines connected in the figure are hard to read. If residual errors are inter-correlated, could you please add a footnote under the figure for simplicity instead of mapping all in the figure?

Response: We agree with Reviewer 2 that Figure 1 was indeed a bit difficult to read. We have deleted all the covariance paths and added a footnote to illustrate it.

Reviewer 3 Report

Topic

The study is very interesting and fills part of the great gap of from family’s hypothesis to research and evidenced through earlier and later assessment of 6 years between two groups of marriges at the same population and country.  The study declined Hypothesis 3, and supported Hypothesis 4 partially 

Abstract

It is clear but was presented very well. 

Authors are encouraged to add these sections

1.     Author mentioned that they excluded some cases that their ages are not assured. But there are not any inclusion or exclusion criteria such as clinical depression, genetics factors of depression or any other disease that cause depression? Other social factors that cause depression and unsatisfaction such spouse infertility, level of education, home stability, etc… 

2.     List the factors and habits that caused high satisfaction and less depression regardless the who is older wife or husband. 

3.     Detection the present barriers that don’t support satisfaction such as what mentioned in comment 1.  

4.     Consider the comments I posted on the manuscript 

Methodology

1.    Effect of both actor and partner was a very excellent method which provide a very realistic result

2.    The study was subjected to china law of marriage ages (female: 20y/ male 22 y) which is a good unified factor follow the country norms and law.

References 

Are adequate but it missed 2021 and 2022 references which is not acceptable 

Author Response

1. Author mentioned that they excluded some cases that their ages are not assured. But there are not any inclusion or exclusion criteria such as clinical depression, genetics factors of depression or any other disease that cause depression? Other social factors that cause depression and unsatisfaction such spouse infertility, level of education, home stability, etc… 

Response: We used couple data from a national representative dataset, and the main purpose of the current study was to probe dyadic influences between Chinese husbands and wives in general. And the CFPS dataset that we used did not have diagnostic information of depression disorder. We could use the CES-D cut-off criteria for depression, but then the accuracy might be questioned. The CFPS dataset did not include question about genetic factors of depression, or infertility, etc., unfortunately. In the preliminary analysis, we found that income was highly correlated with level of education, so we used income as one of the covariate variables in the analysis. We agree with Reviewer 3 that some other influencing factors are worthy of further investigation, and we added this into the limitation section.

2. List the factors and habits that caused high satisfaction and less depression regardless the who is older wife or husband. 

Response: This comment is in line with the first comment. We added a sentence in the introduction regarding the factors related to both life satisfaction and depressive symptoms among married couples (page 2, line 82-86). In the limitation section we mentioned once more that the current study did not account for all the relative factors, and we considered that as one of the limitations (page 13, line 419-421)

3. Detection the present barriers that don’t support satisfaction such as what mentioned in comment 1.  

Response: We think this comment is in line with the first two comments. There were factors that were related to depressive symptoms and life satisfaction but were not accounted for in the current study. We have elaborated this in the limitation section.

References 

Are adequate but it missed 2021 and 2022 references which is not acceptable 

Response: We have added newer references to address this suggestion.